# Rapid invisible frequency tagging (RIFT) does not evoke intermodulation components in the neural response

Alexander Zhigalov[1*], Ole Jensen[2,3]

1 School of Engineering and Innovation, Aston University, Birmingham, United Kingdom, 2 Oxford Centre for Human Brain Activity, University of Oxford, Oxford, United Kingdom, 3 Department of Experimental Psychology, University of Oxford, Oxford, United Kingdom

* a.zhigalov@aston.ac.uk

## Abstract

The human visual system performs nonlinear integrative operations at multiple stages of visual information processing. For instance, integrating parts of visual stimuli into a coherent object involves coordinated neural processing along the visual hierarchy. However, it remains uncertain whether visual integration manifests in a nonlinear neural response, particularly through intermodulation components in the power spectrum. In this study, we used a visual motion paradigm combined with rapid invisible frequency tagging (RIFT) and magnetoencephalography (MEG) to explore nonlinear characteristics of neural responses associated with visual integration. In this paradigm, two grating patches were moving coherently or incoherently, and were modulated by RIFT at 56 and 63 Hz, respectively. The behavioural results revealed that the participants responded more accurately and faster to probes during coherent compared to incoherent motion. Moreover, the type of motion elicited differential effects on pupil dilation, with significantly larger pupil diameter observed during incoherent motion. To evaluate the neural response to coherent and incoherent motion stimuli, we assessed spectral coherence between MEG and RIFT. We observed a strong coherence at the tagging frequencies ($f_1 = 56$ and $f_2 = 63$ Hz) as well as at the higher harmonics at 112 Hz and 126 Hz, respectively. Importantly we did not observe a response at frequencies of the intermodulation ($f_2-f_1$, $f_2+f_1$); nor did we observe a difference when comparing the coherence and incoherent motion. We conclude that in contrast to studies with low-frequency visible tagging, RIFT does not evoke intermodulation components and therefore, its applicability for investigating the neural mechanisms of visual integration might be limited.

## Introduction

The integration of multiple parts of a visual scene into a single object is a fundamental property of the visual system. In this process, spatially separated visual inputs are

**Data availability statement:** MEG data in MATLAB format are available on the OSF server (https://osf.io/a9quw).

**Funding:** Wellcome Trust Discovery Award (grant number 227420) to O.J.; NIHR Oxford Health Biomedical Research Centre (NIHR203316) to O.J. The funders had no role in study design, data collection and analysis, decision to publish, or preparation of the manuscript.

**Competing interests:** The authors have declared that no competing interests exist.

integrated into a coherent percept [1,2]. However, the neural mechanisms underlying visual integration remains unclear, specifically with regards to whether this process involves a nonlinear neural response.

Visual motion paradigms have been used to disentangle the neural response associated with coherent and incoherent motion, and therefore, to gain further insight into the neural mechanisms underlying visual integration. In such paradigms [3], moving parts can be perceived as either a single object or unbound parts. Whereas coherent and incoherent motion of stimuli activates distinct parts of the cortex [4], there is no clear consensus on the spectral characteristics of neural response associated with visual integration [5].

To better understand the spectral characteristics of neural response in perceptual and cognitive tasks, a frequency tagging technique has been introduced, in which rhythmic flashes at specific frequencies (e.g., f1 and f2) are superimposed into visual stimuli [6,7]. The neural response to the stimuli yields complex evoked spectra comprising frequencies related to the periodic tagging as well as intermodulation terms $(n*f1 \pm m*f2)$ which supposedly reflect interactions between specific neural groups [8]. The empirical results support the notion that perceptual integration can be reflected in the intermodulation frequency. Presence of the intermodulation frequencies in the power spectrum also indicates nonlinear characteristics of the visual system since only fundamental frequencies (f1 and f2) and their harmonics (n*f1 and m*f2) can be observed in a linear system [9]. The intermodulation components in the power spectrum have been observed in different paradigms including visual motion, figure-ground segregation, bistable perception and binocular rivalry among others [10]. Interestingly, while EEG studies demonstrate the presence of various combination of intermodulation frequencies [11,12], MEG studies generally report fewer intermodulation components [3,13].

Typically, frequency tagging is applied within the range of 1–30 Hz [14], in part due to technical constraints imposed by screen refresh rates. However, stimulation at such frequencies produces visible flicker which may interfere with task performance and in some cases lead to adverse health effects including the risk of epileptic seizures [15]. In contrast, rapid invisible frequency tagging or RIFT [16,17] utilises high-frequency rhythmic stimuli, usually above 60 Hz, which remain largely imperceptible to participants while still eliciting a robust neural response.

In this study we set out to explore the neural responses reflecting visual integration by combining visual motion paradigm with RIFT and MEG. We hypothesise that visual integration, operationalised through coherent motion of grating patches, is associated with a greater amplitude of intermodulation components compared to incoherent motion. Furthermore, these intermodulation components exhibit distinct spatial patterns across the visual system, which can be effectively captured by MEG.

## Materials and methods

### Participants

Twenty-four participants (mean age: 35; age range: 18–42) with no history of neurological disorders partook in the study. Four participants were excluded from the

analysis due to extensive artifacts. Prior MEG studies [3,9,13] demonstrated that a sample size between 10 and 12 participants is sufficient to reliably detect intermodulation components. The study received approval from the local ethics committee (University of Birmingham, UK) and written informed consent was obtained prior to enrolment. All participants met the standard inclusion criteria for MEG experiments, which included the absence of metal parts in the body and no history of mental illness or disorders. Data collection started on 05/10/2020 and was completed on 10/06/2021. MEG data in MATLAB format are available on the OSF server (https://osf.io/a9quw).

## Experimental paradigm

Two grating stimuli, 5.8° of visual angle, were displayed bilaterally on a projector screen positioned 1.4 meters from the participants' eyes (Fig 1A). One second after the stimuli appeared, the left and right gratings began moving for 2.5 to 3.5 seconds, either in the same direction (coherently) or in opposite directions (incoherently). The direction of motion (up or down) for the left and right gratings was counterbalanced across trials. The gratings moved at a constant speed of 0.5°/s. Participants were instructed to focus on a fixation point and press a button when a cue indicating the direction of motion (coherent or incoherent) appeared at the fixation point. If the direction of motion matched the cue (i.e., equal or unequal sign), participants pressed the button with their middle finger, otherwise, they responded using their index finger. The response instructions, specifying the use of the left or right hand, were counterbalanced among participants.

The luminance of the stimuli was modulated using two RIFT signals, 56 Hz and 63 Hz, (Fig 1B). These frequencies represented a trade-off between minimising the perceptual salience of flicker and maximising the amplitude of the neural response that attenuates with increasing stimulation frequency. Furthermore, the expected 7 Hz intermodulation component, representing the difference between the two tagging frequencies, resides at the boundary between the prominent theta (4–7 Hz) and alpha (8–13 Hz) rhythms, which facilitates its detection by minimising the influence of these endogenous rhythms. We used the PROPixx DLP LED projector (VPixx Technologies Inc., Canada) to present the visual stimuli at a high refresh rate of 1440 Hz and a resolution of 960×600 pixels (see, [18]). The experimental paradigm was implemented in MATLAB (Mathworks Inc., Natick, USA) using Psychophysics Toolbox 3.0.11 [19].

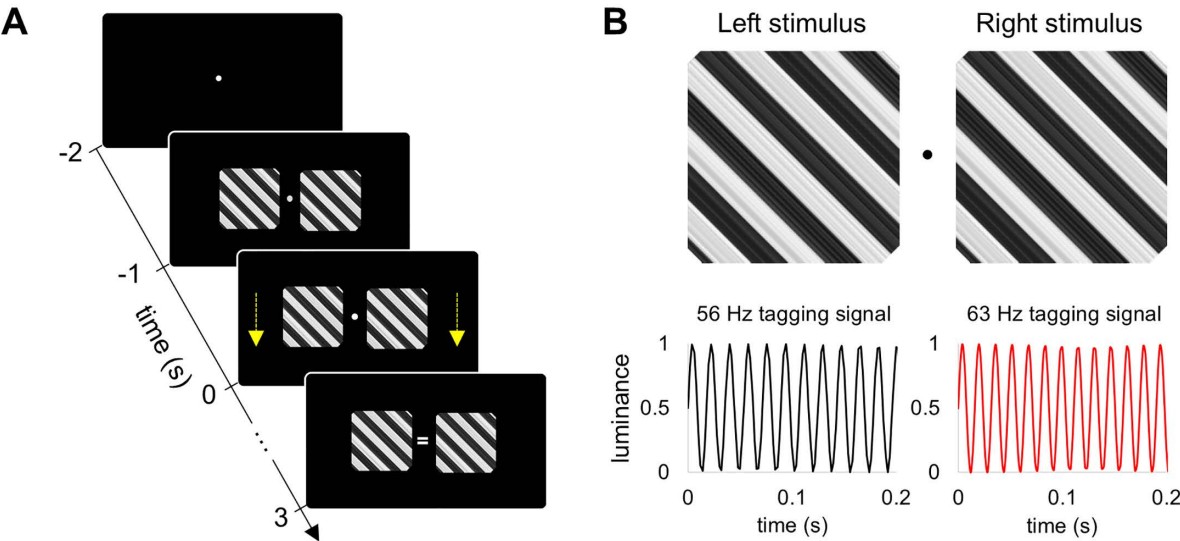

**Fig 1. Experimental paradigm and tagging signals. (A)** Two grating stimuli, exhibiting either coherent or incoherent motion, were presented bilaterally. The motion commenced 1 second after the onset of the flickering stimuli and continued for 2.5 to 3.5 seconds. **(B)** The luminance of the left ("Left stimulus") and right ("Right stimulus") grating stimuli was modulated by two RIFT signals, 56 Hz and 63 Hz.

## MEG and MRI acquisition

Before the measurement, the participant's head position was digitized using the Polhemus Fastrack electromagnetic digitizer system (Polhemus Inc., USA). MEG data were collected using a 306-sensor TRIUX system (MEGIN, Finland) while the participant sat upright at a 60° angle in a dimly lit, magnetically shielded room. The magnetic signals were bandpass filtered between 0.1 and 330 Hz with embedded anti-aliasing filters and sampled at 1000 Hz. Concurrently with the MEG, we recorded the pupil coordinates and diameter using an EyeLink eye-tracker (SR Research, Canada).

The RIFT signals were captured using a custom-made photodetector (Aalto NeuroImaging Centre, Aalto University, Finland) connected to a built-in amplifier in the MEG system. This setup allowed us to acquire the tagging signal together with the MEG data.

Additionally, a high-resolution T1-weighted anatomical images (TR/TE of 7.4/3.5 ms, a flip angle of 7°, FOV of 256 × 256 × 176 mm, 176 sagittal slices, and a voxel size of 1 × 1 × 1 mm3) were acquired using 3-Tesla Phillips Achieva scanner.

## MEG data preprocessing

To reduce external interference and sensor artifacts, we applied signal-space separation (SSS; [20], as implemented in MNE-Python toolbox [21]. SSS decomposes the MEG signal into components originating from inside and outside the measurement volume, discarding the external components associated with environmental noise.

To further reduce artifacts from cardiac signal and eye blinks, we applied independent component analysis (ICA; [22]). Before the analysis, the MEG time series were bandpass filtered between 1 and 100 Hz and downsampled to 250 Hz. Components with topographies corresponding to blinks and cardiac signals were then projected out as follows [23],

$$Y = (I - A(:, r) \cdot W(r, :)) \cdot X$$

where $X$ represents the original MEG data; $A$ and $W$ are the mixing and unmixing matrices, respectively; $I$ is the identity matrix; $r$ denotes the indices of the components corresponding to eye blinks and cardiac signal; and $Y$ represents the MEG data with the artifacts projected out.

The artifact-free MEG time series were segmented into 6-second epochs, ranging from −2–4 seconds relative to the onset of stimulus motion. Although ICA effectively removed blinks from the MEG signal, we also analysed the X-axis and Y-axis channels of the eye-tracker to exclude trials containing blinks and saccades. Eye blinks were detected in the eye-tracker channels by z-scoring the time series and applying a threshold of 5 standard deviations (SD); any deflection above this threshold was classified as a blink. Saccades were identified using a scatter diagram of the X-axis and Y-axis time series from the eye-tracker for each trial. An event was classified as a saccade if the gaze shifted away from the fixation point by 2° and lasted more than 500 milliseconds. Trials contaminated by blinks and saccades were excluded from further analysis. Additionally, we rejected trials containing large-amplitude events (above 5 SD) in the MEG data, which are mainly associated with motion and muscle artifacts. As a result, the number of trials remaining after exclusion was 304 ± 12 (mean ± SD) per participant. For each participant, the number of trials per condition was equalized by randomly selecting the same number of trials.

## Spectral coherence

Spectral coherence between MEG sensors and RIFT signals was calculated using FieldTrip toolbox [23]. Additionally, we generated synthetic tagging signals at 7 and 14 Hz, to evaluate coherence at these intermodulation frequencies which are multiples of the tagging frequency differences (i.e., 56 and 63 Hz),

 

$$C_{xy}(f) = \frac{|P_{xy}(f)|}{\sqrt{P_x(f)P_y(f)}}$$

where $P_{xy}$ is the cross-spectral density estimate of MEG and tagging signal; $P_x$ and $P_y$ are power spectral density estimates of MEG and tagging signals, respectively; $f$ denotes frequency parameter.

$$P_{xy} = F(x)F(y)^*$$

where $F$ denotes Fourier transform and * (asterisk) denotes the complex conjugate.

Spectral coherence was calculated for each trial and then averaged across trials.

### Power spectral density

The power spectral density was calculated using the Fourier transform for each sensor and epoch from −1 to 2.5 seconds relatively to motion onset. To this end, the time series were split into multiple windows of 1 second length and 0.05 second step, weighted by the Hanning taper. The resulting spectral density was averaged within the alpha band (8–13 Hz).

### Source localisation

Spectral coherence between MEG and RIFT was projected onto the source space using the linearly constrained minimum variance (LCMV) beamformer [24], as implemented in Fieldtrip [23]. To construct the forward model, we first segmented the MRI data using FieldTrip, and then the MRI images were manually co-registered with the head shape digitization points obtained via the Polhemus Fastrak system. Finally, a single-shell head model was generated by fitting surface spherical harmonics to the brain surface [25]. The individual anatomy was wrapped to the standard MNI template using non-linear normalisation [23]. A template grid with a 10 mm resolution was applied yielding 3294 source points for each participant. Using the spectral coherence, as described in the preceding section, and the lead field matrices a spatial filter was calculated for each source point.

### Statistical testing

We applied cluster-based permutation test [26] to identify sensors at which neural response to RIFT was above chance level ($p < 0.05$).

The difference between conditions at the group level was evaluated using a paired non-parametric test (two-sided, Wilcoxon signed rank test) with False Discovery Rate correction [27].

## Results

In this study, we presented participants with two grating patches whose luminance was modulated by RIFT signals (56 and 63 Hz) while we simultaneously acquired the MEG data. The motion of the gratings was either coherent or incoherent, which allowed for the dissociation of the neural mechanisms of visual integration. To this end, we calculated spectral coherence between MEG and RIFT signals for both conditions. Additionally, we conducted an exploratory analysis to quantitatively assess the contributions of alpha oscillations and pupil dilation to the visual integration process.

### Behavioural results

To ensure that participants responded correctly to the stimuli, and coherent and incoherent motion patterns were distinguishable, we calculated the percentage of correct responses to coherent and incoherent motion during RIFT (Fig 2A). We found that the accuracy was significantly higher in coherent compared to incoherent motion ($p < 0.0025$, Wilcoxon

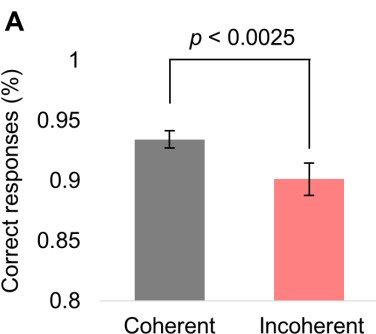
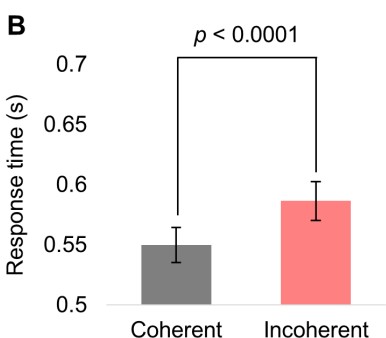

**Fig 2. Behavioural performance. (A)** The percentage of correct responses to coherent and incoherent motion during RIFT. **(B)** The response times to coherent and incoherent motion during RIFT.

test). Similarly, the response times (Fig 2B) were shorter in coherent compared to incoherent motion ($p < 0.0001$, Wilcoxon test).

## Spectral coherence

Spectral coherence was calculated between RIFT and MEG signals. RIFT produced a strong neural response at the tagging frequencies, 56 and 63 Hz, as well as at their harmonics, 112 and 126 Hz, respectively (Fig 3). However, we did not observe intermodulation components peaking at |f2 − f1| or |2*f2 − 2*f1|, as reported elsewhere. The difference in spectral coherence between conditions were not significant ($p > 0.05$ (corrected)) at any frequency.

The coherence at RIFT frequencies (56 Hz and 63 Hz) was largely above the chance level ($p < 0.05$, permutation test) at the occipital, parietal and central sensors (Fig 4). There were several occipital sensors at which coherence was significant at the RIFT harmonics (112 Hz and 126 Hz). However, we did not observe significant changes in coherence at other frequencies, including intermodulation components (7 Hz and 14 Hz). These results demonstrated that in contrast to previous studies with visible tagging, RIFT did not evoke intermodulation components in the power spectrum of neural response.

To gain further insight regarding the anatomical location of the neural sources, we projected spectral coherence into source space using LCMV beamformer approach. The largest values of spectral coherence between MEG and RIFT at 56 and 63 Hz were localised within the primary visual cortex (Fig 5). This result suggested that the neural response to RIFT has limited propagation beyond the primary visual cortex, which is consistent with earlier research [28,29].

## Alpha oscillations

Given that alpha oscillations and neural response to RIFT are linked in perceptual tasks [30] and are both modulated by attention [18], we tested whether the visual integration is reflected in neural response outside of the tagging frequencies, particularly in the alpha band (8–13 Hz). To this end, we calculated the power spectral density in the alpha band during coherent and incoherent motion. There results indicated a slight increase in the alpha power at the occipital sensors during incoherent motion between 0.5 and 1.5 s after motion onset (Fig. 6A). However, the difference between conditions was not significant at the group level ($p > 0.13$; Wilcoxon test). The alpha power tended to increase during incoherent motion, and the largest condition difference was observed in the occipitoparietal areas (Fig 6B).

## Pupil dilation

Considering that motion stimuli may affect neural responses indirectly, for instance, by changing the arousal level, we assessed the changes in pupil diameter at the group level during coherent and incoherent motion. The difference in the

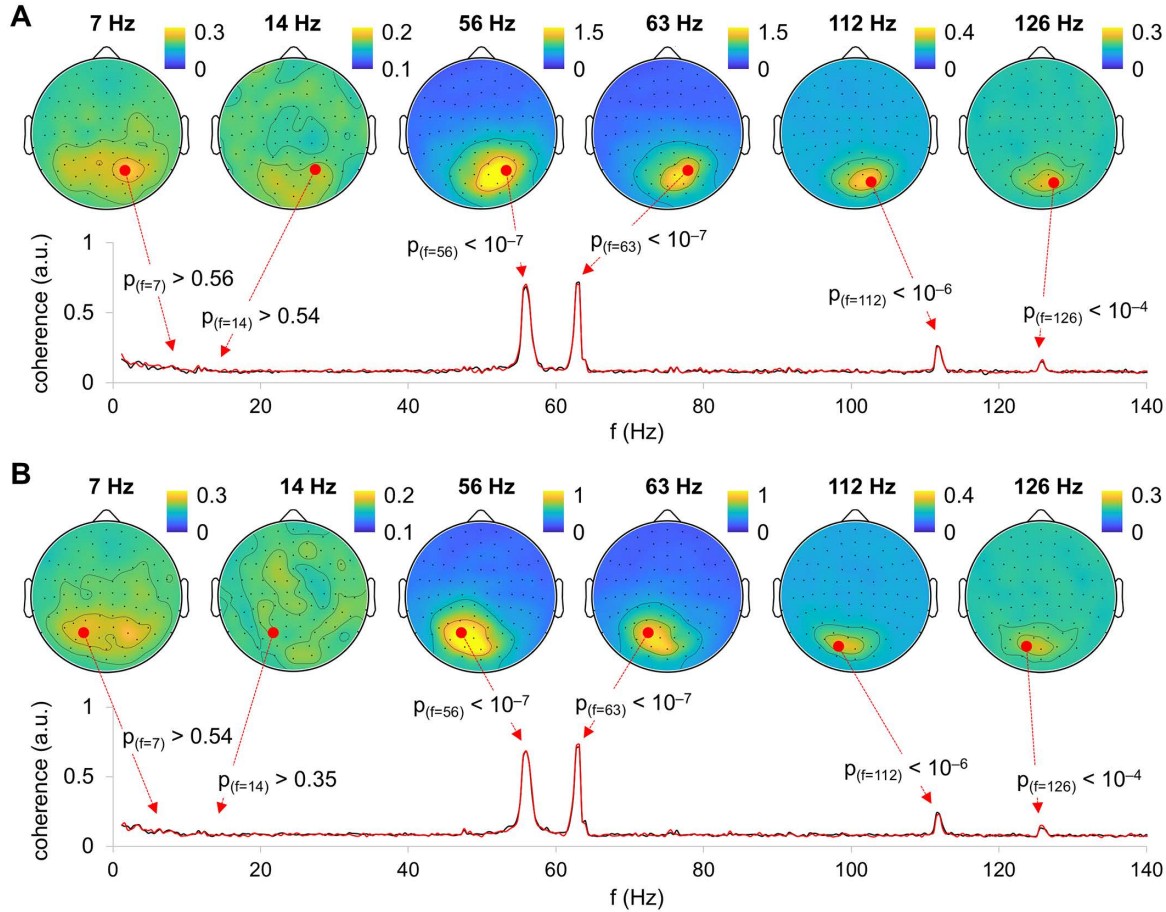

**Fig 3. Spatial patterns of spectral coherence between MEG and RIFT signals.** The coherence was evaluated between MEG and RIFT presented in the left visual hemifield (A) as well as MEG and RIFT presented in the right visual hemifield **(B)**. Black and red lines indicate spectral coherence in coherent and incoherent motion conditions, respectively.

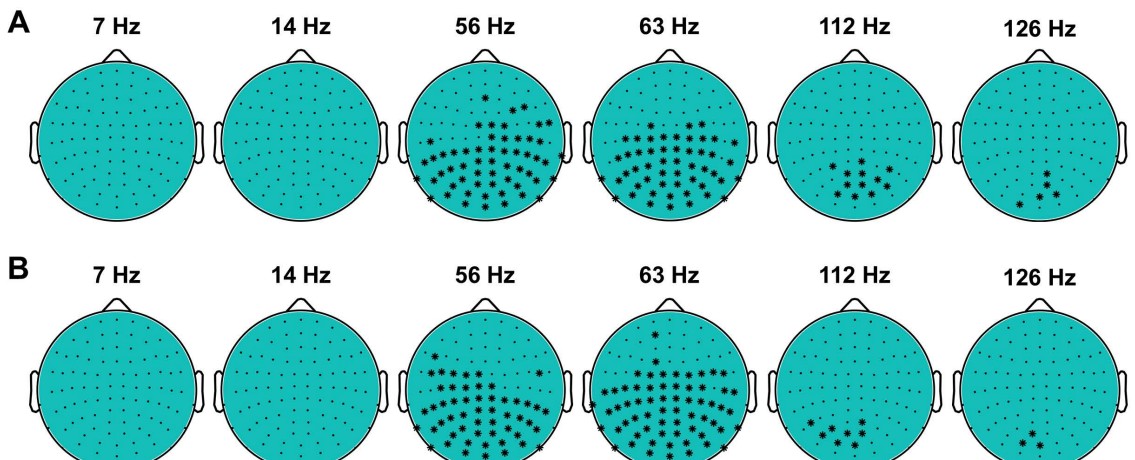

**Fig 4. Statistical analysis of the spatial patterns of coherence.** Topography of the MEG sensors showing significant response at the tagging frequencies of 56 and 63 Hz and their harmonics of 112 and 126 Hz (i.e., 2*f1, 2*f2), but not at the intermodulation frequencies 7 and 14 Hz (i.e., |f2–f1| and |2*f2–2*f1|)). Topographies were calculated for RIFT presented in the left visual hemifield (A) and in the right visual hemifield **(B)**.

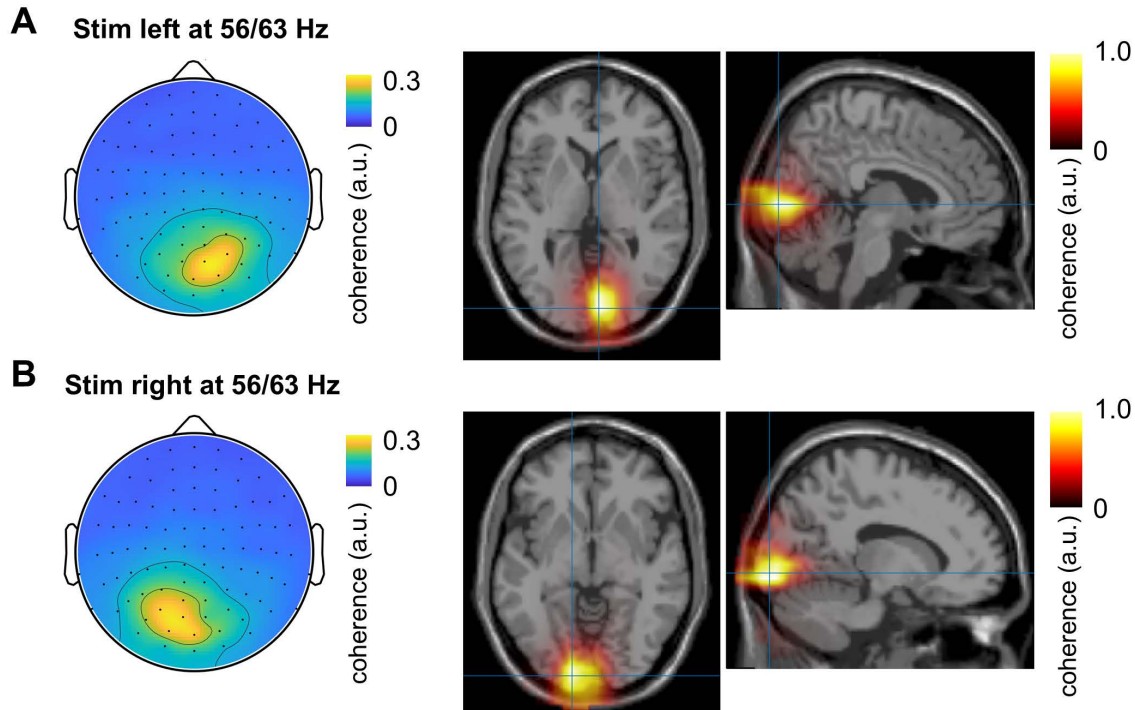

**Fig 5. Source space analysis.** Spectral coherence in sensor and source spaces for RIFT presented in the left hemifield (A) and right hemifield **(B)**, respectively.

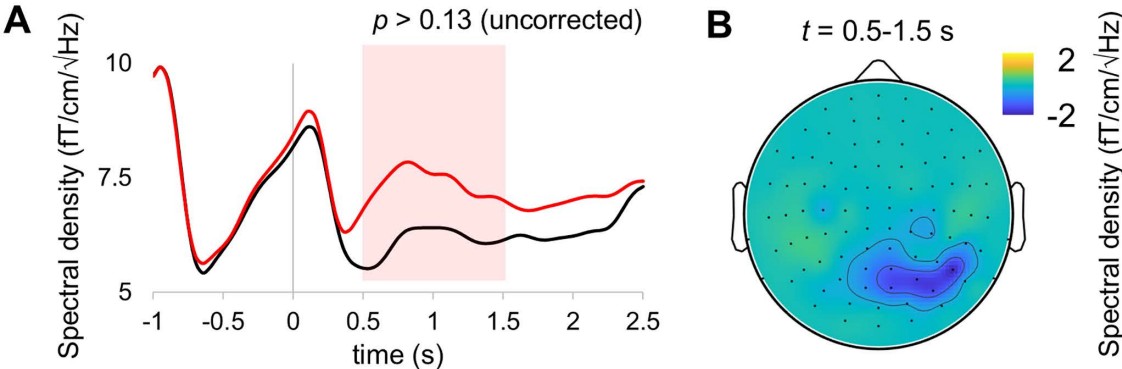

**Fig. 6. Temporal evolution and topography of alpha power during visual motion. (A)** Temporal changes in spectral density in the alpha band during coherent (black line) and incoherent (red line) motion. **(B)** Topography of the condition difference in the spectral density within the 0.5–1.5 interval indicated on panel **A.**

group averaged pupil diameter was significant after approximately 1 s of motion onset (Fig 7A). Interestingly, the pupil diameter was correlated with the alpha power at the group level, and the significant correlation ($p < 0.05$, corrected) was observed in the left centroparietal regions (Fig 7B). This result suggested that coherent and incoherent motion differently influenced the arousal level, which may partly explain the differences in behavioural performance.

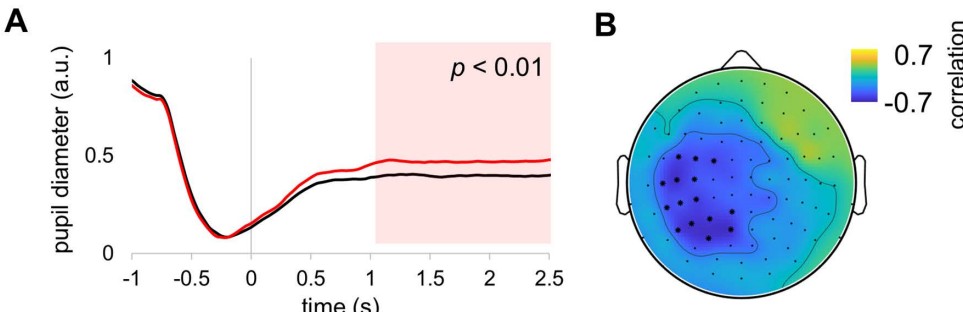

**Fig 7. Pupil dilation and its correlation with alpha power. (A)** Group averaged pupil diameter during coherent (black line) and incoherent (red line) motion. **(B)** Group level correlation between the pupil diameter and alpha power. Sensors with significant correlation are marked with asterisks.

## Discussion

In this study, we investigate the potential of RIFT technique to probe neural mechanisms involved in visual integration. To this end, we presented participants with either coherent or incoherent motion stimuli combined with invisible frequency tagging while simultaneously acquiring MEG.

### Neural response to RIFT and intermodulation components

We found that RIFT evokes a reliable neural response at the tagging frequencies (i.e., f1 and f2) and their harmonics (2*f1 and 2*f2), but we did not observe intermodulation components at |f2 – f1|. This result is consistent with our earlier study utilising RIFT at 63 and 78 Hz [16], where intermodulation components were absent in response to static images. The absence of intermodulation components may be attributable to several factors. Firstly, given that these components typically exhibit relatively low amplitudes in MEG recordings [9], a larger sample size may be required to detect statistically significant effects. Secondly, the local nature of neural response to invisible flicker suggests limited propagation of high-frequency activity beyond the primary visual cortex [28,29] and given that visual integration is likely to occur at higher levels of the visual hierarchy, the integration effect is not reflected in the observed neural response. Thirdly, intermodulation components |f2 – f1| are typically observed at lower frequencies around 1.26 Hz [31] and 2.5 Hz [13], and therefore, the 7 Hz difference between the two tagging frequencies used in the current study might be insufficient to elicit intermodulation components.

### Neural response to coherent and incoherent motion

Neural response to coherent and incoherent motion was evaluated through spectral coherence calculated between RIFT and MEG. The coherence was not significantly different between these conditions at any of the tagging frequencies, their harmonics or intermodulation components. This result may suggest that moving gratings did not engage visual integration rather other processes such as attention, resulting in distinct behavioural outcomes during coherent versus incoherent motion.

To test whether spatial attention was employed in this paradigm, we evaluated changes in alpha power associated with coherent and incoherent motion. Although alpha power tends to increase in occipital areas during incoherent motion, this effect was not significant. Nevertheless, this may suggest that motion stimuli influenced attention or arousal levels.

### Neurophysiological factors affecting neural response to motion

Given that pupil dilation reflects physical characteristics of visual stimuli [32] as well as neurophysiological characteristics such as arousal level [33], we used eye-tracking data to explore factors that may influence the neural response. Our

results showed that the pupil diameter became significantly larger during incoherent motion after approximately 1 s of motion onset. Since the luminance of the stimuli was identical in both conditions, this effect is unlikely to be explained by stimuli characteristics alone. On the other hand, and in line with our behavioural data showing a significant decrease in accuracy and response time during incoherent motion, we can assume that motion stimuli do affect the arousal level.

It has been shown that periods of elevated arousal are coincide with increased attention [33] and pupil-linked arousal can be reflected in a negative correlation between pupil diameter and alpha power [34]. To test this, we assessed the correlation between alpha power and pupil diameter within 1–2.5 s after motion onset, and found a negative correlation, primarily in the left centroparietal regions. This result further confirmed that motion stimuli in our paradigm modulate the arousal level but may not activate the mechanism of visual integration.

## Conclusion

In this study, we demonstrated that invisible frequency tagging evokes robust neural response at the tagging frequencies and their harmonics but does not summon intermodulation components. Therefore, compared to low-frequency visible tagging, the use of RIFT for investigating neural mechanisms of visual integration may be limited. Furthermore, exploratory analyses of alpha power and pupil dilation suggested that moving gratings may primarily modulate arousal levels rather than engage neural mechanisms of visual integration, thereby limiting their suitability for studying visual integration.

## Acknowledgments

We thank Jonathan L. Winter for providing help with the MEG recordings.

## Author contributions

**Conceptualization:** Alexander Zhigalov, Ole Jensen.

**Data curation:** Alexander Zhigalov.

**Formal analysis:** Alexander Zhigalov.

**Funding acquisition:** Ole Jensen.

**Investigation:** Alexander Zhigalov, Ole Jensen.

**Methodology:** Alexander Zhigalov, Ole Jensen.

**Project administration:** Ole Jensen.

**Software:** Alexander Zhigalov.

**Supervision:** Ole Jensen.

**Visualization:** Alexander Zhigalov.

**Writing – original draft:** Alexander Zhigalov.

**Writing – review & editing:** Alexander Zhigalov, Ole Jensen.

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
