## [Decision Letter · Decision Letter 0]

17 Oct 2025

Dear Dr. Zhigalov,

We look forward to receiving your revised manuscript.

Kind regards,

Valentina Bruno

Academic Editor

PLOS ONE

[Wellcome Trust Discovery Award (grant number 227420) to O.J.; NIHR Oxford Health Biomedical Research Centre (NIHR203316) to O.J.].

Additional Editor Comments (if provided):

Reviewers' comments:

Reviewer's Responses to Questions

**Comments to the Author**

1. Is the manuscript technically sound, and do the data support the conclusions?

Reviewer #1: No

Reviewer #2: Yes

2. Has the statistical analysis been performed appropriately and rigorously?

Reviewer #1: Yes

Reviewer #2: Yes

3. Have the authors made all data underlying the findings in their manuscript fully available?

Reviewer #1: Yes

Reviewer #2: Yes

4. Is the manuscript presented in an intelligible fashion and written in standard English?

Reviewer #1: Yes

Reviewer #2: Yes

Reviewer #1: The research question is interesting, but I have major concerns that I outline below.

- Reading the introduction, it remains unclear whether the main goal of the study is to investigate visual binding through coherent vs incoherent motion or to specifically test for intermodulation frequencies as a marker of visual integration. At present, both aspects are discussed in parallel, which makes it difficult to identify the primary research question.

- The introduction does not fully explain why rapid invisible frequency tagging (rather than classical frequency tagging) is necessary for this paradigm. Is RIFT expected to yield stronger or more reliable intermodulation responses in MEG?

- The stated hypothesis appears only at the very end of the introduction. Bringing this forward and linking it more explicitly to the cited studies would strengthen the logical flow.

Furthermore, the text introduces visual binding in terms of coherent vs incoherent motion but does not specify clearly how this is measured or defined in the current paradigm. A clearer link between the perceptual concept and the neural measure would help.

- Given that the study reports null results, it would be important to include a power analysis of the sample size. Intermodulation frequencies are typically small in amplitude, and the absence of significant effects may therefore be explained by insufficient statistical power rather than by the absence of a true neural effect. Providing an a priori or post hoc power analysis would help the reader to interpret the null findings more appropriately.

- While behavioral data clearly differentiate coherent and incoherent motion, the MEG data do not. This raises the question of whether the chosen paradigm (RIFT + gratings) is sensitive enough to capture binding-related neural processes at all. The authors should discuss whether the lack of neural differentiation reflects methodological limitations rather than theoretical conclusions. The absence of intermodulation frequency is not the only null result because also coherence did not differentiate between conditions.

- The Discussion tends to interpret the absence of intermodulation frequencies as evidence that RIFT primarily drives local responses in V1. However, without independent confirmation (e.g., source localization), this explanation remains speculative. Alternative explanations (e.g., insufficient power, suboptimal tagging frequencies, or signal cancellation in MEG) should be considered more explicitly.

- The rationale for selecting exactly these two tagging frequencies is not provided. Were they chosen because of hardware constraints, or because they are optimal for eliciting nonlinear interactions? The choice is crucial given that intermodulation effects are frequency-dependent.

- The exploration of alpha oscillations feels underdeveloped. The authors report a non-significant trend but then interpret it as potentially meaningful. This risks over-interpretation. Either this section should be framed more cautiously or complemented with stronger analysis.

- The pupil effects are interesting but remain peripheral to the central hypothesis of visual binding. At present, this section feels like a compensatory analysis for null MEG results. The authors should clarify whether pupil and alpha analyses were preregistered hypotheses or exploratory.

- The stated aim is to investigate neural correlates of visual binding using intermodulation components. Since neither intermodulation nor MEG differences between coherent and incoherent motion were observed, the conclusions should be more restrained. Currently, the manuscript risks overstating the significance of secondary findings (pupil diameter, alpha trends).

Reviewer #2: The manuscript by Zhigalov and Jensen is clear and well written. Methods are sound.

Major comments

All hypotheses are well defined, except for the Alpha analysis, which appears somewhat post hoc and out of context (the results for the Alpha are also only at the level of a trend). Thus I wonder whether this section should be included at all.

In the Discussion section, it is not addressed to what extent the difference between the two tagging frequencies (56 and 63 Hz) might be insufficient to elicit intermodulation frequencies, nor how the chosen frequency range relates to previous work in the same line of research.

**Do you want your identity to be public for this peer review?** For information about this choice, including consent withdrawal, please see our Privacy Policy

Reviewer #1: No

Reviewer #2: No

---

## [Author Response · Author response to Decision Letter 1]

31 Oct 2025

Academic Editor

• The files are now named according to the journal style requirements.

[Wellcome Trust Discovery Award (grant number 227420) to O.J.; NIHR Oxford Health Biomedical Research Centre (NIHR203316) to O.J.].

If this statement is not correct you must amend it as needed. Please include this amended Role of Funder statement in your cover letter; we will change the online submission form on your behalf.

• The statement regarding the role of the funders has now been included in both the Acknowledgement section and in the online submission form as follows, “The funders had no role in study design, data collection and analysis, decision to publish, or preparation of the manuscript” (lines 353–355).

• All data are now available on the OSF server, https://osf.io/a9quw. The statement on data availability is now included in the Materials and Methods section as follows, “MEG data in MATLAB format are available on the OSF server (https://osf.io/a9quw)” (lines 96–97).

• In the original submission, figure captions were embedded within the manuscript text, directly after the paragraph in which they are first cited. We now applied the appropriate style (i.e., bold font instead of italic) for figure captions in accordance with PLOS One formatting guidelines.

• No recommendations were made by the reviewers regarding citations of specific works.

----

We thank reviewers for their constructive comments, and we did our best to address these points. The changes made to the original manuscript are highlighted in red (see, “Revised Manuscript with Track Changes.docx”).

----

Reviewer #1: The research question is interesting, but I have major concerns that I outline below.

- Reading the introduction, it remains unclear whether the main goal of the study is to investigate visual binding through coherent vs incoherent motion or to specifically test for intermodulation frequencies as a marker of visual integration. At present, both aspects are discussed in parallel, which makes it difficult to identify the primary research question.

• Indeed, the primary goal of this study was to examine intermodulation components as indicators of visual integration.

• We did our best to clarify this aspect throughout the text.

- The introduction does not fully explain why rapid invisible frequency tagging (rather than classical frequency tagging) is necessary for this paradigm. Is RIFT expected to yield stronger or more reliable intermodulation responses in MEG?

• We have now added further clarification regarding RIFT in the text as follows, “Typically, frequency tagging is applied within the range of 1–30 Hz (14), in part due to technical constraints imposed by screen refresh rates. However, stimulation at such frequencies produces visible flicker which may interfere with task performance and in some cases lead to adverse health effects including the risk of epileptic seizures (15). In contrast, rapid invisible frequency tagging or RIFT (16, 17) utilises high-frequency rhythmic stimuli, usually above 60 Hz, which remain largely imperceptible to participants while still eliciting a robust neural response” (lines 72–78).

- The stated hypothesis appears only at the very end of the introduction. Bringing this forward and linking it more explicitly to the cited studies would strengthen the logical flow.

Furthermore, the text introduces visual binding in terms of coherent vs incoherent motion but does not specify clearly how this is measured or defined in the current paradigm. A clearer link between the perceptual concept and the neural measure would help.

• We now restructured and rewrote the Introduction section to improve the logical flow.

• Specifically, in the first paragraph, we emphasised that “the neural mechanisms underlying visual integration remains unclear, specifically with regards to whether this process involves a nonlinear neural response” (lines 48–50).

• We subsequently described the visual motion paradigm as an approach to investigating visual integration, and frequency tagging as a technique to explore the spectral characteristics of neural activity underlying visual integration.

• We then elaborated on the advantages of RIFT over conventional frequency tagging, and stated hypothesis of our study as follows, “We hypothesise that visual integration, operationalised through coherent motion of grating patches, is associated with a greater amplitude of intermodulation components compared to incoherent motion. Furthermore, these intermodulation components exhibit distinct spatial patterns across the visual system, which can be effectively captured by MEG” (lines 80–84).

- Given that the study reports null results, it would be important to include a power analysis of the sample size. Intermodulation frequencies are typically small in amplitude, and the absence of significant effects may therefore be explained by insufficient statistical power rather than by the absence of a true neural effect. Providing an a priori or post hoc power analysis would help the reader to interpret the null findings more appropriately.

• We have now provided a justification for the sample size, drawing on evidence from previous studies. The following text has been added, “Prior MEG studies (3, 9, 13) demonstrated that a sample size between 10 and 12 participants is sufficient to reliably detect intermodulation components” (lines 90–91).

• We also addressed this point in the Discussion section, “Firstly, given that these components typically exhibit relatively low amplitudes in MEG recordings (9), a larger sample size may be required to detect statistically significant effects” (lines 305–307).

- While behavioral data clearly differentiate coherent and incoherent motion, the MEG data do not. This raises the question of whether the chosen paradigm (RIFT + gratings) is sensitive enough to capture binding-related neural processes at all. The authors should discuss whether the lack of neural differentiation reflects methodological limitations rather than theoretical conclusions. The absence of intermodulation frequency is not the only null result because also coherence did not differentiate between conditions.

• In fact, the intermodulation components have been observed for both static and moving stimuli in previous studies, therefore, we expected to observe intermodulation regardless the presence of moving gradings. In this study, we specifically tested whether the type of motion (i.e., coherent or incoherent) changes the amplitude of intermodulation component.

• In the Discussion section, we have now elaborated on the factors that could account for the absence of intermodulation components, “Firstly, given that these components typically exhibit relatively low amplitudes in MEG recordings (9), a larger sample size may be required to detect statistically significant effects. Secondly, the local nature of neural response to invisible flicker suggests limited propagation of high-frequency activity beyond the primary visual cortex (28, 29) and given that visual integration is likely to occur at higher levels of the visual hierarchy, the integration effect is not reflected in the observed neural response. Thirdly, intermodulation components |f2 – f1| are typically observed at lower frequencies around 1.26 Hz (31) and 2.5 Hz (13), and therefore, the 7 Hz difference between the two tagging frequencies used in the current study might be insufficient to elicit intermodulation components” (lines 305–314).

- The Discussion tends to interpret the absence of intermodulation frequencies as evidence that RIFT primarily drives local responses in V1. However, without independent confirmation (e.g., source localization), this explanation remains speculative. Alternative explanations (e.g., insufficient power, suboptimal tagging frequencies, or signal cancellation in MEG) should be considered more explicitly.

• We have now conducted the source localisation of spectral coherence. The methodology is detailed in the Materials and Methods section (lines 190–200), and the key findings are outlined in the Results section (lines 254–259). The results further confirmed that the neural response in localised in the primary visual cortex (see, Fig 5).

• In the Discussion section, we now provided alternative explanations of the absence of intermodulation (lines 305–314).

- The rationale for selecting exactly these two tagging frequencies is not provided. Were they chosen because of hardware constraints, or because they are optimal for eliciting nonlinear interactions? The choice is crucial given that intermodulation effects are frequency-dependent.

• This clarification has now been provided in the text, “These frequencies represented a trade-off between minimising the perceptual salience of flicker and maximising the amplitude of the neural response that attenuates with increasing stimulation frequency. Furthermore, the expected 7 Hz intermodulation component, representing the difference between the two tagging frequencies, resides at the boundary between the prominent theta (4–7 Hz) and alpha (8–13 Hz) rhythms, which facilitates its detection by minimising the influence of these endogenous rhythms” (lines 119–124).

- The exploration of alpha oscillations feels underdeveloped. The authors report a non-significant trend but then interpret it as potentially meaningful. This risks over-interpretation. Either this section should be framed more cautiously or complemented with stronger analysis.

• This section has now been rewritten as follows, “Neural response to coherent and incoherent motion was evaluated through spectral coherence calculated between RIFT and MEG. The coherence was not significantly different between these conditions at any of the tagging frequencies, their harmonics or intermodulation components. This result may suggest that moving gratings did not engage visual integration rather other processes such as attention, resulting in distinct behavioural outcomes during coherent versus incoherent motion. To test whether spatial attention was employed in this paradigm, we evaluated changes in alpha power associated with coherent and incoherent motion. Although alpha power tends to increase in occipital areas during incoherent motion, this effect was not significant. Nevertheless, this may suggest that motion stimuli influenced attention or arousal levels” (lines 316–325).

- The pupil effects are interesting but remain peripheral to the central hypothesis of visual binding. At present, this section feels like a compensatory analysis for null MEG results. The authors should clarify whether pupil and alpha analyses were preregistered hypotheses or exploratory.

• We now clarified that the analyses related to pupil dilation are exploratory and intended to further investigate factors that may account for the absence of intermodulation components.

• This section has been rewritten as follows, “Given that pupil dilation reflects physical characteristics of visual stimuli (32) as well as neurophysiological characteristics such as arousal level (33), we used eye-tracking data to explore factors that may influence the neural response. Our results showed that the pupil diameter became significantly larger during incoherent motion after approximately 1 s of motion onset. Since the luminance of the stimuli was identical in both conditions, this effect is unlikely to be explained by stimuli characteristics alone. On the other hand, and in line with our behavioural data showing a significant decrease in accuracy and response time during incoherent motion, we can assume that motion stimuli do affect the arousal level. It has been shown that periods of elevated arousal are coincide with increased attention (33) and pupil-linked arousal can be reflected in a negative correlation between pupil diameter and alpha power (34). To test this, we assessed the correlation between alpha power and pupil diameter within 1–2.5 s after motion onset, and found a negative correlation, primarily in the left centroparietal regions. This result further confirmed that motion stimuli in our paradigm modulate the arousal level but may not activate the mechanism of visual integration” (lines 327–340).

- The stated aim is to investigate neural correlates of visual binding using intermodulation components. Since neither intermodulation nor MEG differences between coherent and incoherent motion were observed, the conclusions should be more restrained. Currently, the manuscript risks overstating the significance of secondary findings (pupil diameter, alpha trends).

• We agree with the reviewer, and reframed the conclusion as follows, “In this study, we demonstrated that invisible frequency tagging evokes robust neural response at the tagging frequencies and their harmonics but does not summon intermodulation components. Therefore, compared to low-frequency visible tagging, the use of RIFT for investigating neural mechanisms of visual integration may be limited. Furthermore, exploratory analyses of alpha power and pupil dilation suggested that moving gratings may primarily modulate arousal levels rather than engage neural mechanisms of visual integration, thereby limiting their suitability for studying visual integration” (lines 343–349).

Reviewer #2: The manuscript by Zhigalov and Jensen is clear and well written. Methods are sound.

Major comments

All hypotheses are well defined, except for the Alpha analysis, which appears somewhat post hoc and out of context (the results for the Alpha are also only at the level of a trend). Thus I wonder whether this section should be included at all.

• We agree that the analysis appears as post hoc. It has now been clarified that the analyses related to alpha oscillations are exploratory and aimed to investigate factors accounting for the absence of intermodulation components.

• This section has now been rewritten as follows, “Neural response to coherent and incoherent motion was evaluated through spectral coherence calculated between RIFT and MEG. The coherence was not significantly different between these conditions at any of the tagging frequencies, their harmonics or intermodulation components. This result may suggest that moving gratings did not engage visual integration rather other processes suc

---

## [Decision Letter · Decision Letter 1]

13 Feb 2026

Rapid invisible frequency tagging (RIFT) does not evoke intermodulation components in the neural response

PONE-D-25-45066R1

Dear Dr. Zhigalov,

We’re pleased to inform you that your manuscript has been judged scientifically suitable for publication and will be formally accepted for publication once it meets all outstanding technical requirements.

Kind regards,

Kiyoshi Nakahara, PhD

Academic Editor

PLOS One

Additional Editor Comments (optional):

Reviewers' comments:

Reviewer's Responses to Questions

**Comments to the Author**

Reviewer #1: All comments have been addressed

2. Is the manuscript technically sound, and do the data support the conclusions?

Reviewer #1: Yes

3. Has the statistical analysis been performed appropriately and rigorously?

Reviewer #1: Yes

4. Have the authors made all data underlying the findings in their manuscript fully available?

Reviewer #1: Yes

5. Is the manuscript presented in an intelligible fashion and written in standard English?

Reviewer #1: Yes

Reviewer #1: I thank the authors that have answered to all points and I believe the paper is now ready for publication

**Do you want your identity to be public for this peer review?** For information about this choice, including consent withdrawal, please see our Privacy Policy

Reviewer #1: No

---

## [Editor Report · Acceptance letter]

PONE-D-25-45066R1

PLOS One

Dear Dr. Zhigalov,

I'm pleased to inform you that your manuscript has been deemed suitable for publication in PLOS One. Congratulations! Your manuscript is now being handed over to our production team.

Kind regards,

on behalf of

Dr. Kiyoshi Nakahara

Academic Editor

PLOS One